# Travel to Mars-like Places on Earth: A New Branch of Sustainable Ecotourism in Lut Desert World Heritage Site, Iran

**Amir Ghorbani** [1], **Ali Zangiabadi** [2,*], **Hossein Mousazadeh** [3,*], **Farahnaz Akbarzadeh Almani** [4], **Kai Zhu** [5] **and Lóránt Dénes Dávid** [6,7,*]

1   Department of Tourism, Faculty of Geographical Sciences and Planning, University of Isfahan, Isfahan 81746-73441, Iran; amir1.tourism@gmail.com
2   Department of Geography and Urban Planning, Faculty of Geographical Sciences and Planning, University of Isfahan, Isfahan 81746-73441, Iran
3   Department of Regional Science, Faculty of Science, Eötvös Loránd University, 1053 Budapest, Hungary
4   Department of Tourism Management, Budapest Business School, University of Applied Sciences, 1149 Budapest, Hungary; farahnaz.akbarzadeh.48@unibge.hu
5   Faculty of Resources and Environmental Science, Hubei University, Wuhan 430062, China; hizhukai@163.com
6   Faculty of Economics and Business, John von Neumann University, 6000 Kecskemet, Hungary
7   Institute of Rural Development and Sustainable Economy, Hungarian University of Agriculture and Life Sciences, 2100 Godollo, Hungary
*   Correspondence: a.zangiabadi@geo.ui.ac.ir (A.Z.); hmosazadeh5575@yahoo.com (H.M.); dr.david.lorant@gmail.com (L.D.D.)

**Abstract:** Traveling to space and walking on other planets has always been a great dream for many tourists. Given that space tourism is not available to everyone, adventurers have always been looking for special and strange places that evoke the feeling of traveling to other planets, especially Mars. One of these places which is very similar to Mars is the Lut Desert World Heritage Site. The present study aims to introduce the Martian sites of the Lut Desert and offers a special type of trip to this beautiful desert that can further preserve it with a sustainable development approach. The statistical sample of the research is based on the qualitative analysis method, consisting of 18 participants, which consists of experts, desert tour guides, and tourists who have visited the studied sites. After the data collection process, the interviews were transcribed and analyzed using Maxqda 2020 software. The results of the research indicate that the four sites identified in the research, according to the participants, are similar to the images and videos published of Mars. Moreover, this new branch of desert ecotourism can develop sustainable ecotourism in the Lut Desert.

**Keywords:** tourism management; space tourism; Mars-like places; sustainable ecotourism; Lut Desert

## 1. Introduction

Ever since the images of Mars were released, humans have sought to experience being on Mars. The sand dunes, the Yardangs, and the rocks of Mars were examples of these attractions [1]. However, space travel is not available to all enthusiasts, and some researchers have claimed that the environmental conditions of Mars are not compatible with the human body in many ways [2]. Therefore, tourists have searched for places like space and Mars on Earth. Adventure tourists refer to several places as Earth Martians in their travel reports, including Devon Island in Canada, the Atacama Desert in Chile, and Wadi Rum in Jordan [3–5]. Recently, the Lut Desert World Heritage Site (LDWHS) has also been considered as one of these sites. Comparing published images from the surface of Mars with the LDWHS emphasizes the similarity between them [6]. Even now, a new form of travel package to the LDWHS, such as walking on Mars on Earth and moonlight hike tours, has been designed for this purpose by DWHS specialized guides for tourists. LDWHS is jointly located in three provinces of Sistan and Baluchestan, Kerman, and South Khorasan in Iran [7]. One of the most important reasons for the registration of the beautiful

LDWHS in the UNESCO World Heritage has been its special morphology [8]. Due to the shape of its geological structures, its weather conditions, its high sand dunes, and the strange human feeling in the LDWHS, many professional tourists consider the desirability of traveling to the LDWHS equivalent to the desirability of the planet Mars. Expert guides of the LDWHS believe that the path from Earth to Mars is achieved by walking on the LDWHS. The 25 km walk between Kaluts and also from the crater of Prof. Kardavani and Prof Mahmoudi in Ravar County to the south side of Gandom Beryan show Mars on the Earth for tourists. Kaluts are among the most wonderful geomorphic features of the LDWHS [9]. While in classical ecotourism, traveling to the depths of the desert by car is more desirable, sustainable ecotourists prefer to immerse themselves in more relaxed thoughts during walking tours in the LDWHS with fantasy titles such as walking on Mars and moonlight hike tours. Moreover, such tours reduce the damage to the desert texture because there are no cars. In addition, tourists' behaviors in sustainable tourism are pro-environmental behaviors and include actions that protect the environment or minimize the negative effects of human activities on the environment [10–12]. Furthermore, Georgakopoulou and Delitheou [13] acknowledged that sustainable tourism development should be promoted through the development of alternative forms of tourism in the form of respect for the environment and sustainable ecotourism standards. Therefore, LDWHS tourists say that they travel to similar destinations such as the LDWHS instead of traveling to desired but inaccessible destinations, such as Mars. However, only limited research has explored the relationship between travel to a similar place and the main destination [14]. On the other hand, many researchers such as Ghorbani et al. [15], Maghsoudi et al. [16], Zabihi et al. [17], and Negahban and Roshan [18] have studied the LDWHS from the perspective of geotourism, geology, etc. Therefore, the present study, using the opinion of tourists and specialized LDWHS guides, introduces the most similar parts of the LDWHS to Mars and compares the images of the LDWHS and Mars from the visual point of view and the view of the participants. Finally, the data are analyzed using a theme analysis method. An attempt was made to present a new perspective according to tourism needs in the future. The findings of this study are expected to pave the way for future research and help tourism specialists, researchers, and policymakers to better plan tourism management. Moreover, the findings of this study will be of great interest to the readers, particularly those involved in the tourism industry and sustainable ecotourism. The present study was conducted with the official permission of the LDWHS World Heritage Site base. In this research, we attempt to investigate the similar potentials and attractions between Mars and the LDWHS as an alternative destination based on understanding tourists' attitudes.

## 2. Literature Review

### 2.1. Alternative Destination Studies

The approach of traveling to a place similar to a tourist destination that is not accessible has been discussed many times in the literature of tourism management and destination studies, and accordingly, various titles have been created for tourist destinations [19]. However, the focus of the research literature in previous studies has been on comparing the two destinations on the Earth. Gentile [20] described an innovative approach to finding similar places, especially in the field of tourism destination suggestions. After understanding how the users refer to or represent a place, they would try to find a similar one. Eckert and Pechlaner [21] suggested the development of alternative products as a new approach to sustainable tourism in the designated destination. Harb and Bassil [22] showed that bilateral tourism flows are not only specified by agents affecting the attractiveness of the destination, but also by the attractiveness of alternative destinations. This means that alternative destinations for tourists should have the charms and wonders of the original location. Furthermore, Georgakopoulou and Delitheou [13] acknowledged that sustainable tourism development should be promoted through the development of alternative forms of tourism in the form of respect for the environment and sustainable ecotourism standards. Moreover, alternative research (e.g., Schieber et al. [23] and Schindler et al. [24]) on tourist



destinations has studied their theoretical foundations and literature, but few scientific studies, such as Slavin's [25], have been conducted to compare the capabilities of traveling to other planets on the Earth.

### 2.2. LDWHS Tourism

Desert tourism, an exciting branch of ecotourism, has grown a lot in the last several years [26]. Repeatedly, when tourists have traveled to the heart of the LDWHS, they have addressed their strange feelings about the LDWHS on social media and linked it to the special features of Mars; in tourism management literature, that is called the impact of place on the tourists [27]. The LDWHS has been attracting more tourists, and recent years have witnessed the popularity of desert tourism as a modern tourism product in the new area [28]. Tourists travel to the deserts for various purposes such as enjoying the sunset in the desert; Night Sky Tourism (NST), or astronomical tourism (the program consists of night-time starwatching with telescopes, multimedia night programs, and solar viewing during the day); wildlife photography; paragliding; camping; sandboarding; and camel riding [29]. However, in recent years, one of the biggest changes in sustainable ecotourism was linked to the LDWHS. Tourists claim that when they are in the LDWHS, they feel that they are not on the planet. In recent years, many tourists have traveled to LDWHS to discover more of the wonders of the LDWHS and compare them to similar phenomena on Mars. Furthermore, one of the passages to the LDWHS in Nehbandan in South Khorasan Province passes to Shahdad in Kerman through a mountain range called the Martian Mountains, which is very similar to Mars [15]. Therefore, it is necessary to introduce the values of the LDWHS to global tourism [16,17]. In a part of the LDWHS near the sandy Everest, there are sand dunes similar to the zebras that surround a Calligonum shrub, where tourists view it as an environmental war to survive. The planned development of tourism in the LDWHS will have numerous benefits such as conservation; environmental education activities, providing local benefits and economic development; and finally, responsible movement and footprint management [30]. Figure 1 shows the LDWHS ecotourism potentials:

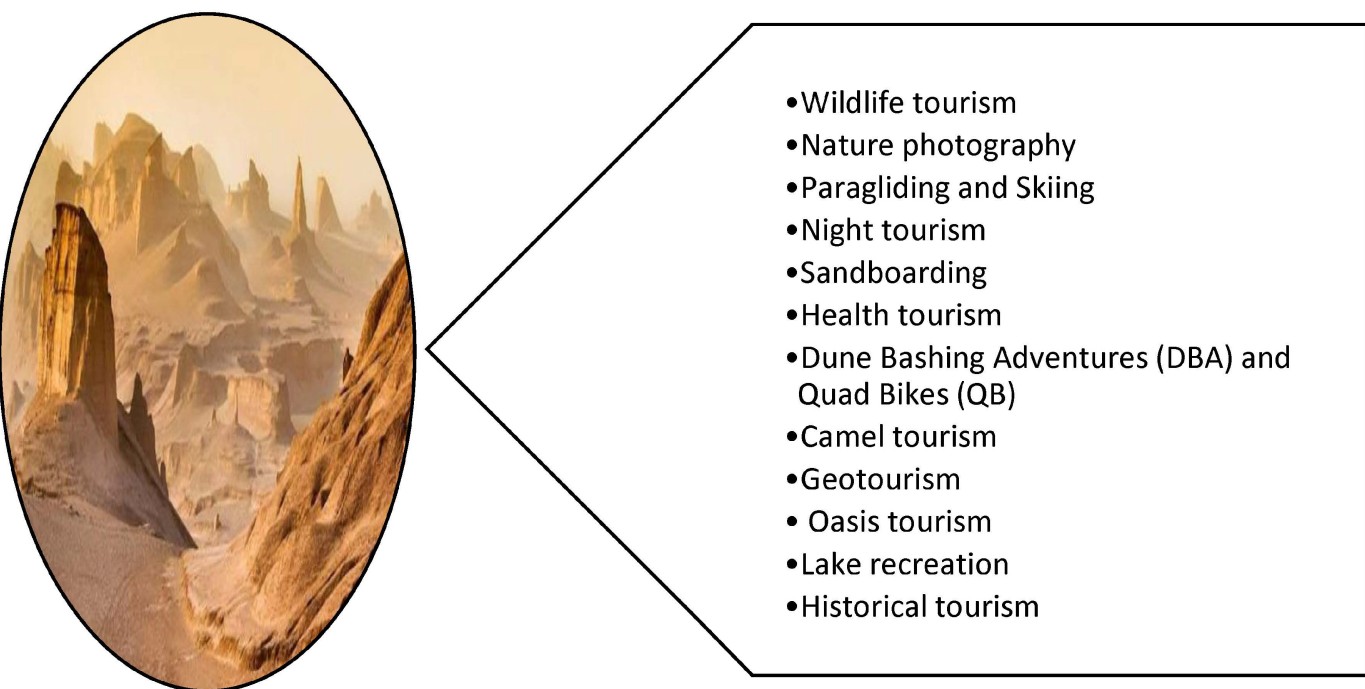

- Wildlife tourism
- Nature photography
- Paragliding and Skiing
- Night tourism
- Sandboarding
- Health tourism
- Dune Bashing Adventures (DBA) and Quad Bikes (QB)
- Camel tourism
- Geotourism
- Oasis tourism
- Lake recreation
- Historical tourism

**Figure 1.** LDWHS potentials (source: research findings).

### 2.3. Sustainable Ecotourism

In recent decades, it has been proven that ecotourism has failed to fulfill its duties to preserve nature [30]. To restart tourism in the current situation and turn tourism into a platform for overcoming pandemics, sustainable ecotourism is a good platform. Sustainability has changed the interests of tourists. Sustainable ecotourism has a much more precise approach to nature conservation than ecotourism, while experts consider ecotourism as a modern way to destroy nature [31]. In sustainable ecotourism, tours are made by very small groups that are concerned with nature conservation, and the impulse for travel is different. Traveling to pristine nature away from the hustle and bustle with a very small number of tourists, following health guidelines to prevent the spread of COVID-19, and adhering to the policies of the sustainable environment is a platform to restart tourism in the COVID-19 era that is linked to the principles of sustainable tourism [32]. For example, tourists may travel to a place distant from the Earth by imagination [33]. Sustainable ecotourism seeks to create a relaxed environment, and instead of crowded travel, there are groups of less than five travelers [34]. This approach is now implemented by some LDWHS guides in the form of travel packages, and tourists enjoy their trip with minimal damage to the desert texture.

## 3. Methodology

The method of the present study is an open and friendly interview with tourists while holding a moonlight hike tour in the LDWHS, and the method of data analysis is qualitative. This method is an interpretive approach that avoids numbers and instead attaches importance to cultural and natural facts [35]. In the present study, considering the specificity of the subject, the grounded theory is an appropriate method that is one of the most widely used qualitative research methodologies [36]. MAXQDA-2020 software is used for qualitative data analysis. Grounded theory is an inductive and exploratory research method that allows researchers in various subject areas to develop a theory and proposition instead of relying on existing and pre-formulated theories [37]. Grounded theory is a method that focuses more on the participant's experience, and its founders, Barney Glaser and Anselm Strauss, describe grounded theory as "discovering data theory" [38]. This approach has been supported by many researchers because of its ability to provide in-depth information on phenomena [39]. Furthermore, when we have no knowledge about phenomena or little knowledge about a phenomenon and its process; the grounded theory method is an appropriate method [40]. Grounded theory allows the research question to be examined from a novel perspective and has the power to examine the subject as perceived by the participants [41]. In the present study, tourists who participated in the LDWHS moonlight hike tour were used as participants, the researchers interviewed them openly during the tour, and the tourists expressed their feelings about the LDWHS Martian sites. In addition, experts and two specialized tour guides of the LDWHS also expressed their views.

### 3.1. Study Area

The LDWHS was registered as Iran's first natural heritage site at the 40th UNESCO World Heritage Summit in July 2016 as No. 1505 on the World Heritage List. The LDWHS is located in the east of Iran, and the three provinces of South Khorasan, Kerman, and Sistan and Baluchestan contain part of the LDWHS (Please see Figure 2). The LSWHS is an area of 80,000 km$^2$ between the Nehbandan fault in the east and the Nayband fault in the west. In parallel to each other, these two faults, with a distance of about 200 km, create the width of the LDWHS (LDWHS base, 2020). Figure 3 shows the collective statistics of LDWHS domestic and foreign tourists from 2015 to 2020:

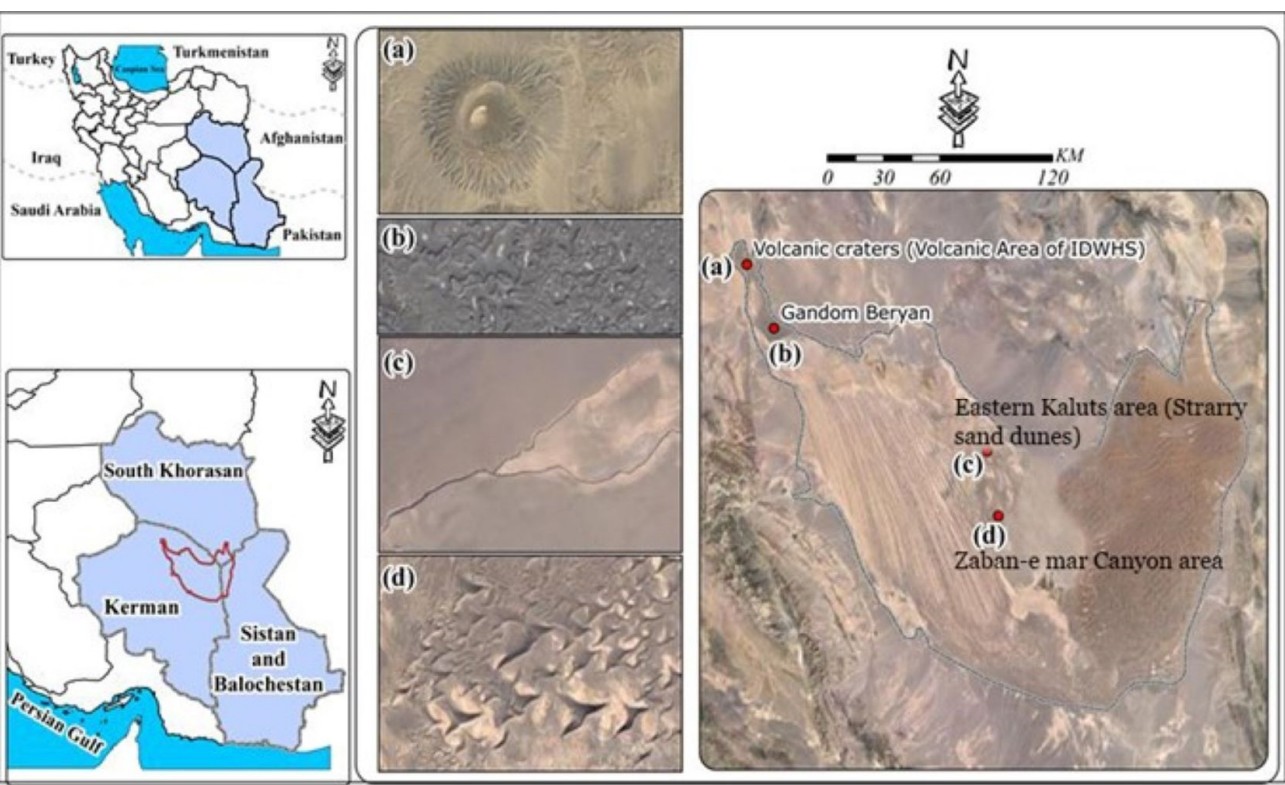

**Figure 2.** Lut Desert and 4 most Mars-like places. (**a**–**d**) they are study sites that's in this study introduced as Mars-like Places on Lut desert in Iran.

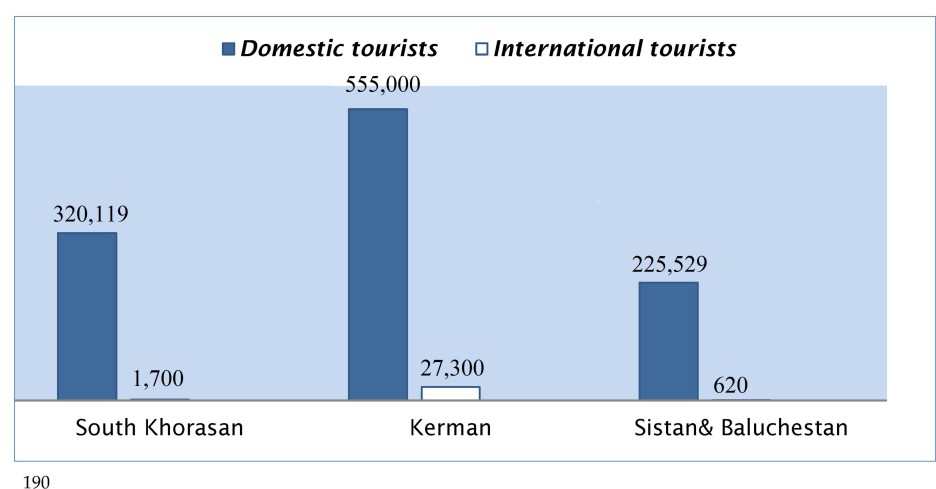

190

**Figure 3.** Collective statistics of LDWHS tourists from 2015 to 2020 (LDWHS Base, 2020).

*3.2. Data Collection*

According to the World Register of LDWHS as Iran's First Natural Heritage at the 40th UNESCO World Heritage Summit, researchers first raised the suggested topic with the LDWHS base. Researchers were asked to submit their research project in order to issue a formal research permit in the form of a proposal to review the subject and assess its adequacy. The issue was discussed in a formal meeting, and while the innovation aspect was confirmed by the working group, the researchers were officially allowed to start working in the LDWHS. The study was approved by the BL99006 code, and permission for the study was granted to the researchers. Then, with an official letter from the director of the LDWHS research, the researchers were allowed to do research, interviews, and

photography in the area. Participants, as a statistical sample of the study, included thirteen tourists who participated in the LDWHS moonlight hike tour, two LDWHS tour guides from the private sector, and the LDWHS expert in the Ministry of Cultural Heritage, Handicrafts (Table 1). The researchers traveled with the tourists during the tour and the interview was conducted in an open and friendly manner. The reason for choosing the members of the moonlight hike tour as participants in the research is that this type of trip aims to understand the illusion of the desert in the moonlight of the LDWHS at midnight and reach one of the LDWHS Martian sites at sunrise. At the start of the tour, at midnight on foot, and at the Martian site, tourists were asked to express their feelings. Due to the fact that the moonlight hike tour is a "special interests" tour and tourists have to walk for hours in the desert, the number of tourists is limited, and the researchers were able to interview all the participants during the tour. The interviews were recorded simultaneously using a sound recorder, and participants were aware of this. Researchers posed open questions, and individuals responded in a friendly manner. Each interview lasted about ten to fifteen minutes, during which the questions were asked (Appendix A). Researchers have previously tested and confirmed the quality of the questions' number, type, and subject matter.

**Table 1.** Participants (source: research findings).

| Row | Section | Type | Number |
|:---:|:---:|:---:|:---:|
| 1 | Private | Moonlight hike tourists | 13 |
| 2 | Private | LDWHS tour guide | 2 |
| 3 | Public | LDWHS director | 1 |
| 4 | Public | LDWHS expert | 2 |
| | Total | | 18 |

Finally, the raw recorded interviews were processed according to the following process (Figure 4):

*Transcribing of interviews*

- Most important basic steps in content analysis.
- The researchers are tasked with reading the answers meticulously and distinguishing whether each sentence is related to their research questions (Kyngäs, 2020).
- If possible, it is best to send the transcribed text to the interviewee for approval(Quinn et al., 2020; Lan, 2020).

*Determining the theme and sub-theme*

- After determining the themes, it is determined which theme the sub-theme is related to (Lan, 2020 ).

*Data Coding*

- This stage is one of the most important stages of grounded theory, which requires the active participation of researchers
- Whereas open coding divides the general data of the interview into related categories, investigating the relationship and interaction of these categories is done by axial coding, which leads to larger categories (Morabi Jouybari et al., 2023).

**Figure 4.** Data processing (source: research findings) [42–45].

### 3.3. Reliability and Validity in Content Analysis

3.3.1. Reliability Analysis

In the present study, Holsti's coefficient of reliability was used to measure the validity of the obtained results. For this purpose, the data were categorized and coded by another researcher. The researcher extracted the themes without knowing the results of the first

stage of analysis, and then the results were compared. If the similarity coefficient is more than 70%, reliability is confirmed. The similarity percentage of the results is determined using the Holsti's coefficient formula:

$$PAO = 2\,M/(N1 + N2) \tag{1}$$

where:

PAO: Percentage of agreement between two coders.
M: Common codes.
N1: Codes of the first investigator.
N2: Codes of second investigators [46].
Calculate percentage:

$$PAO = \frac{2 \times 76}{98 + 88} = 89.551 \tag{2}$$

### 3.3.2. Validity

To measure the validity of the research, the expert survey method was used. For this purpose, the research model and the results of the research were given to the experts in form of 7-point Likert scale ranges, and they declared their level of agreement with the results. Expert survey is one of the best ways to check the quality of research results in content analysis [42].

## 4. Results

### 4.1. Extracted Theme and Sub-Themes

In this section, based on the analysis of the transcribed interviews, the main and sub-themes are identified. After identifying the main codes, sub-themes are assigned to the corresponding group. In content analysis, the analysis of the interviews continues until theoretical saturation, which means when the analyst decides that no new themes are being extracted [46]. Table 2 contains the MAXQDA analysis results:

**Table 2.** Themes and sub-themes.

| Themes | Sub-Themes | Frequency |
|---|---|---|
| Climate | Extreme temperature fluctuations | 14 |
| | Abundant wind, strong dust storms, and surface vibrations | 10 |
| | Strange, scary, and at the same time, beautiful rain | 10 |
| Kaluts | The imaginary city of the LDWHS and its special shape | 18 |
| | One of the most similar structures in the LDWHS to the Mars surface | 16 |
| | Illusion, liberation, and silence | 14 |
| | Walking in the corridors of the LDWHS and a red sky in the sunset | 13 |
| Valleys and Rocks | The corridors of the snake tongue valley and the pass through it | 11 |
| | The special form of erosion of valleys, especially the snake tongue valley | 7 |
| | The special structure and rocks | 6 |
| Volcanic area | Illusion, liberation, and silence | 9 |
| | The special color of volcanic cone's area | 8 |
| Sand dunes | View of the LDWHS from the top of the sand dunes | 18 |
| | The height of the sand dunes | 16 |
| | The special shape of the sand dunes | 8 |
| Accessibility | Travel to the LDWHS requires special equipment | 9 |
| | Routing and mapping skills | 9 |
| | Traveling to the LDWHS, like Mars, requires special training | 8 |
| Geomorphology | The LDWHS has a high potential area for finding meteorites (a piece of space) | 11 |
| | Specific sedimentary forms | 9 |
| | Soil types in both areas have many similarities in terms of color | 8 |
| | Due to the special geomorphological conditions of the two areas, the formation of floods seems very interesting | 7 |
| | Mars mountain in the LDWHS margin | 6 |

*4.2. Similarity Framework*

Based on the participants' feedback and a comparative study, the common features of the LDWHS and Mars were listed (Figure 5):

## Sand dunes

- Gabriel, 1938
- Alavipanah et al.,2007
- Yazdi et al.,2014
- Roback et al., 2020
- Dundas, 2020
- Stillman et al.,2020

## Volcanic area

- Javidi Moghaddam et al., 2020
- Schindler et al, 2019

## Yardang

- Lyons et al, 2020
- Yuan et al., 2020
- Ding et al, 2020

## Dust Storm and wind

- Zhang et al., 2006
- Guzewich et al, 2019
- Cantor et al, 2019
- *Nasa*
- Radebaugh et al.,2017
- Ghodsi, 2017
- Schieber et al, 2020
- Favaro et al, 2020

## Climate change

- Winkler, 2020
- Williams, 2014
- Nerozzi and Holt, 2019

## Rocks & valley

- Golombek et al 2020
- Kite, 2019

**Figure 5.** Common features of the LDWHS and Mars [9,23,24,47–65].

## 5. Discussion

Ecotourists now consider being on Mars one of the motivations for their trip to the LDWHS, and instead of having fun doing things that damage the environment of the LD-WHS, these tourists are more sensitive to the preservation of the natural environment [31]. Moreover, in the natural development scenario, all land-use changes are permitted [47]. Therefore, the present study examined the similarity of the LDWHS and Mars from the participants' point of view. This was a qualitative and abstract study of which main criterion was the imagination of the participants. According to Table 2 and the content analysis

results, seven themes (i.e., Climate, Kaluts, Valleys and Rocks, Volcanic area, Sand dunes, Accessibility, and Geomorphology) make them feel like they are on Mars.

### 5.1. LDWHS Climate Conditions

The LDWHS is one of the hottest deserts on our planet with extreme fluctuations in temperature over a given day [66]. The LDWHS receives < 30 mm of precipitation annually. The average yearly temperature is 27.5 °C, and the average minimum and maximum daily air temperatures of the warmest and coldest months of the year are −2.6 °C and 50.4 °C, respectively [67]. The LDWHS, is classified in the Global Bioclimatic Classification (GBC) as 'tropical hyper desertic' and the hottest and driest bioclimate in the world [68]. A new measurement through the sampling campaign (4 March 2017) returned the surface temperature as high as 78.2 °C in Yalan dunes [48]. Participants acknowledged during the interview that the LDWHS has its own weather conditions that are different from a normal place. Temperature differentials between night and day combine with frequent wind, strong dust storms, vibrations of the ground, and terrifying rains that produce red floods. Furthermore, according to Kass et al. [69], "the largest of dust storms on Mars are Global Dust Events (GDE) that affect essentially every aspect of the Martian atmosphere but do not occur in every Mars Year". These reasons for this similarity were mentioned during the interview. However, according to Figure 6, Mars' weather conditions are also quite unstable.

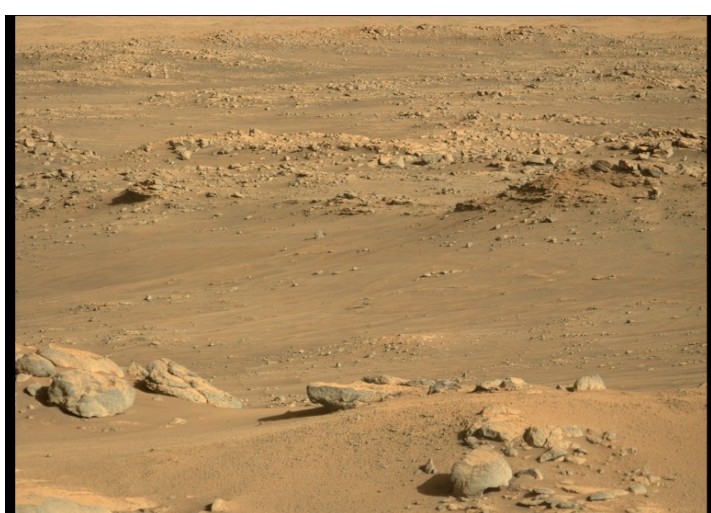 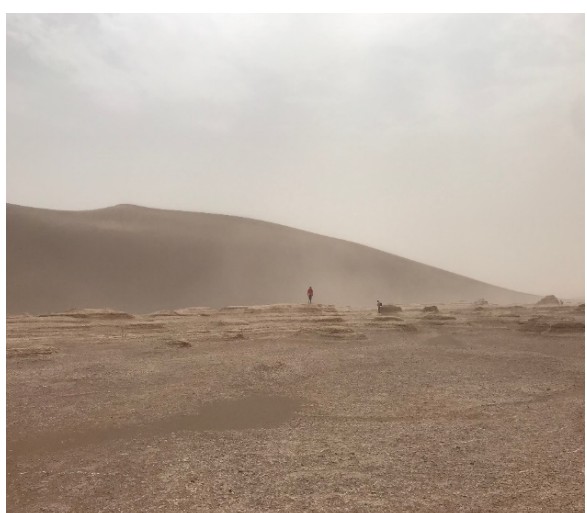

**Figure 6.** Climate conditions in LDWHS and Mars (source: research findings).

"The temperature difference in the LDWHS is very high. During the night, when we camped in the LDWHS to observe the night sky, we lit a fire due to the extreme cold. This was while we could not walk in the desert in the heat of the day. This climate in the LDWHS is very similar to what I read about the Mars climate, especially when we see sandstorms, strong dust storms, surface vibrations, and scary rains during the day," (Participant 7, moonlight hike tourists)

"Extreme temperature fluctuation of the LDWHS is one of the attractions that make me travel to Iran every year. When I am in the LDWHS, away from the hustle and bustle of life and without access to communication, especially at sunset, I feel that I have really gone beyond the planet Earth and it seems that I am on another planet like Mars," (Participant 13)

*5.2. The Imaginary City of KaLDWHSs*

The western part of the LDWHS is characterized by KaLDWHSs (also known as Yardangs). KaLDWHSs are formations created by wind abrasion and deflation [68]. The special shape of KaLDWHSs, walks in the corridors of the KaLDWHSs, and a red sky in the sunset, combined with a sense of illusion, liberation, and silence, have turned the area into a terrestrial Mars for tourists. However, according to Figures 7 and 8 and NASA studies, there are also yardangs similar to the Shahdad KaLDWHSs on Mars. Ding et al. [70] showed that based on the principles of the Earth-based yardang progression, it is possible to recognize the particular level of development and examine the geomorphological processes as well as the climatic background for yardangs on Mars. Studies show that Yardang landforms on Mars are concentrated in the equatorial region of the Medusae Fossae Formation (MFF) [71], and compositions of fluvial/lacustrine sediments also exist on Mars [49,72]. Many tourists who visit the LDWHS KaLDWHSs for the first time without any background think that these KaLDWHSs are the remnants of an ancient civilization or city. Thus, it is easy to assume that these rock masses were created by humans from the flat desert surface. Among European tourists, however, this is not uncommon. However, the truth is not so, as the KaLDWHSs are completely natural (Figure 7).

> "As a leader of foreign tourists in the LDWHS, I have seen many times that foreign tourists, seeing the KaLDWHSs of Shahdad, think that these structures are remnants of urbanization in the region. With the advancement of science and technology, new phenomena have been identified from Mars. On several recent trips, two American tourists traveling to the LDWHS claimed that the Mars yardangs were very similar to those of the Shahdad KaLDWHSs. When they returned to the hotel, their accommodation center, they showed me pictures of the Mars yardangs, which was really amazing to me," (Participant 9, LDWHS international leader)

> "According to previous experiences to see the beauties of the LDWHS at night, I participated in a moonlight hike tour. The sunset in the Lut Desert is considered one of the most romantic natural landscapes in the whole world. With the sunset, it is as if the world is coming to an end. The sky turns red and the desert sets foot in another world. As if you are on another planet or universe. If you are looking for the best place to watch the sunset, I suggest you get to the top of Shahdad," (Participant 4, moonlight hike tourists)

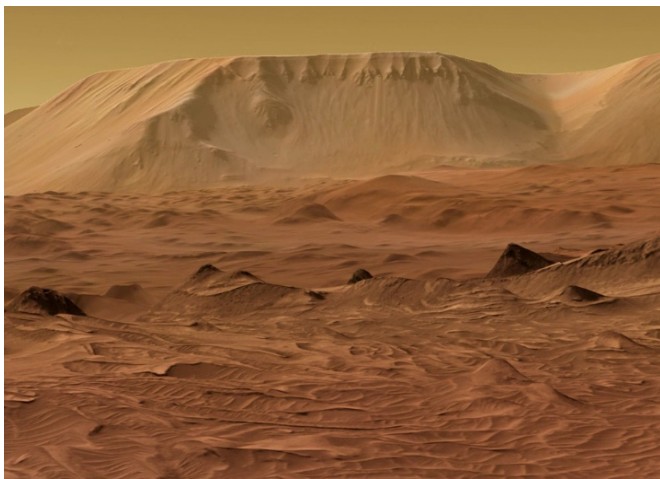 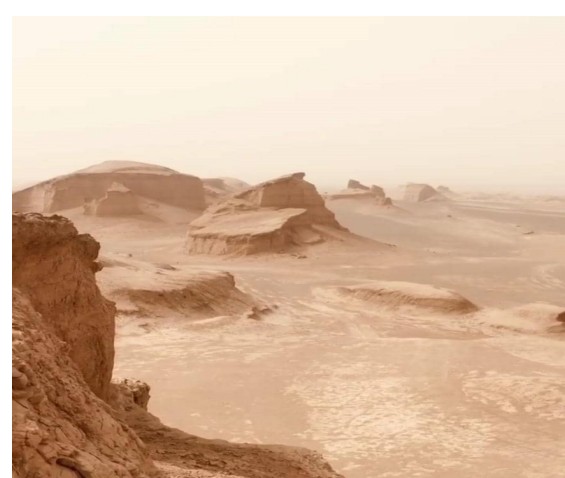

**Figure 7.** Kaluts in LDWHS and Mars (source: research findings).

> "While travel to Mars may not be available to everyone, the spectacular geological features of the LDWHS and yardangs (KaLDWHSs) are very similar to those photos taken by NASA on Mars. The spectacular KaLDWHSs of the LDWHS can

provide hours of fun and ideal photography opportunities for tourists to really feel like they have traveled to another planet. Watching these sights at sunset or sunrise is a Martian-like experience for the tourist, which will definitely not be possible to watch anywhere else," (Participant 5)

### 5.3. LDWHS Valleys and Rocks

The basement of the LDWHS contains pre-Jurassic metamorphic rocks (pelitic schist) as well as Jurassic sediments intruded by younger (Jurassic to Tertiary) granitic pLDWHSons and covered by Tertiary mafic and felsic volcanics [73–75]. The LDWHS has a high potential for meteorite preservation [76]. In the LDWHS, there are strange rocks and valleys that are the result of severe water and wind erosion [18]. During the interview, participants said that the specific shape of the valleys and rocks in the LDWHS, especially the snake tongue valley in Kerman Province, corresponds to their mental perception of Mars (Figure 8).

"The snake tongue valley is one of the most amazing areas in the LDWHS, which is very famous in many ways, especially for the type of rock. The spectacular view of the snake tongue valley, which is a combination of sandy and rocky hills and dry desert plains, can bring imaginative Mars landscapes in front of the observer. What gives this spectacular landscape an abstract, imaginary feature are the sharp cuts in the rock and wall rocks of the snake tongue valley as if made from a Mars pattern," (Participant 5)

"As we know, most of what scientists know about volcanoes and Mars includes information gathered from Martian meteorites found on Earth. The LDWHS is one of the centers where similar meteorites found elsewhere can be found. A few years ago, we set out for a scientific study in the LDWHS and sampled soil and rock samples from LDWHS. All my colleagues and companions believed that the special shape of the valleys and rocks in the LDWHS is similar to the images and specimens that NASA received from the surface of Mars," (Participant 2)

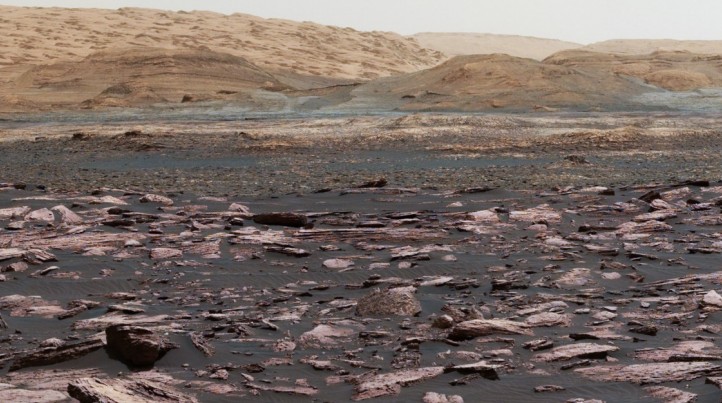 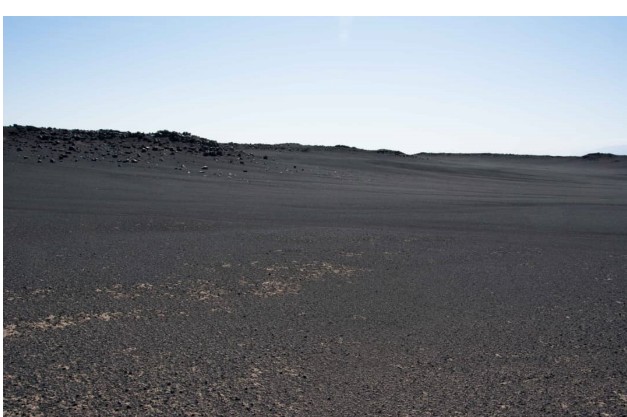

**Figure 8.** Valleys and rocks in LDWHS and Mars (source: research findings).

### 5.4. Volcanic Area

The LDWHS is a tectonically active area with many active faults [77]. Mars is likely to be home to some of the oldest volcanoes in the solar system. New evidence suggests that Mars has witnessed volcanic activity for at least two billion years. A small Martian meteorite found on the African continent in 2012 is evidence of this claim. Recently, in the latest information from Mars, the characteristics of Mars' volcanic activities have been studied in many scientific studies [78,79]. At the level of the LDWHS block, there are about 40 Quaternary volcanic cones. Volcanoes in the northern part of the LDWHS may be the result of the eruption of the Afghan block beneath the LDWHS block, and the volcanoes on the southern fringe of the LDWHS are part of the Magma arc of the Makran eruption

zone [80]. The volcanic area of the LDWHS is one of the beautiful attractions of this area in the field of geotourism [81]. NASA studies have also discussed Mars' volcanic activities according to Figure 6. The "Gandom Briyanak" area in the LDWHS, one of the hottest places in the world, is one of the volcanic areas, which tourists call Black Mars (Figure 9).

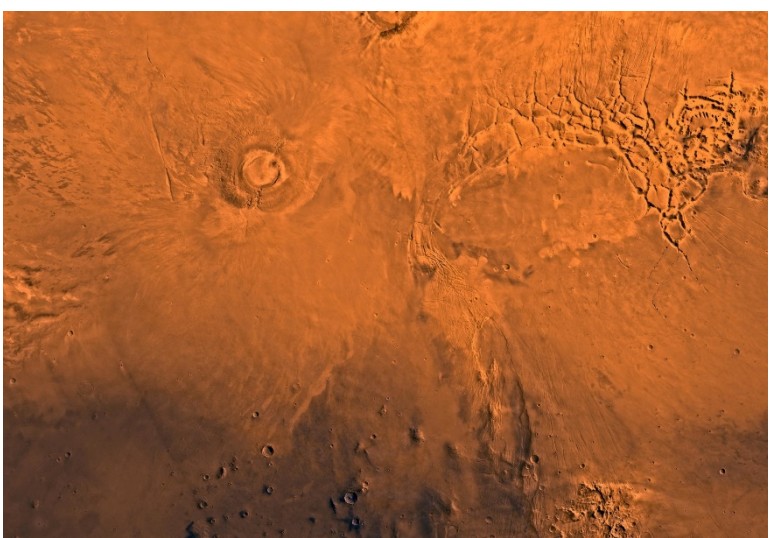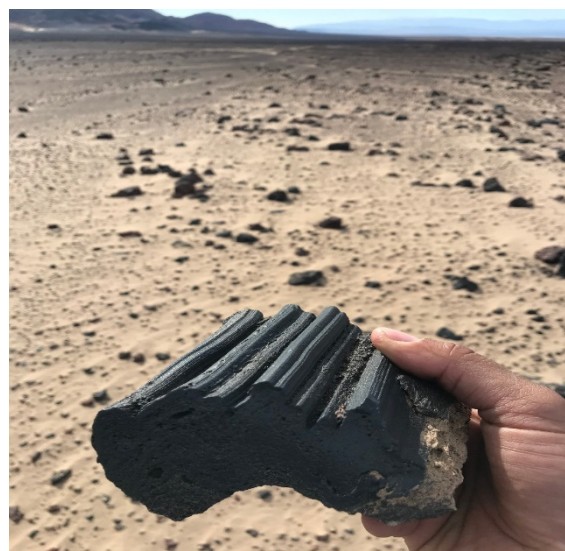

**Figure 9.** Volcanic area in LDWHS and Mars (source: research findings).

> "Volcanoes have often been a threat to human life, but the LDWHS volcanoes are a major source of tourism. The volcano in the Gandom Briyanak region in the LDWHS is shown more in films made about Mars. The tranquility and quietness of this area make tourists feel like they are on another planet such as Mars," (Participant 11). "The special color of the volcanic cone's area in the Lut Desert is exactly the same as the ones I have seen on the surface of Mars on astronaut websites, especially on NASA website," (Participant 4)

*5.5. Sand Dunes*

The sand dunes, also called "Sandy Everest", are located in the LDWHS in South Khorasan and Sistan and Baluchestan provinces and are approximately 500 m above sea level [15]. Furthermore, "Rig-e Yalan" is located in the east of the central LDWHS and is the intersection of the three provinces of South Khorasan, Kerman, and Sistan and Baluchestan. This region is one of the pristine deserts that amazes tourists with the highest sand dunes in the world. About 10 km after Nehbandan towards Shahdad, mountains and sand dunes appear on both sides of the road, which are known as miniature or Martian mountains. These are beautiful sedimentary mountains, and due to rapid erosion against wind and rain, they have become serrated and full of edges and frequent cracks. Tourists and locals also refer to them as "Mars" mountains because of their unusual shape [80]. The LDWHS, in terms of notable features, has lots of potential and exceptional global conditions, including the very high sand dunes, the highest and longest yardangs (Kaluts) [82], and nebkhas [16]. NASA researchers have also discussed the sand dunes of Mars. Specialists tracked the movement of nearly 500 individual dunes on Mars, all using data assembled by NASA's Mars Reconnaissance Orbiter. Participants consider these sand dunes as one of the most Martian parts of the world (Figure 10).

> "On the one hand, the cone-shaped Martian mountains with that imaginative grey color, and on the other hand, the height of the sand dunes on its boundary in the LDWHS have been able to depict a beautiful natural attraction and the resemblance of Mars on the earth for tourists," (Participant 14)

"The high sand dunes of the LDWHS amaze any viewer. The sand dunes are one of the most prominent attractions of the LDWHS, which provide beautiful scenery for tourists. The most important sand dunes of the LDWHS that most tourists like to see and walk on when traveling to LDWHS are Sandy Everest and Rig-e Yalan. When you look at the desert from the sand dunes, as far as the eye can see, you see only the very high sand dunes that surround you," (Participant 3)

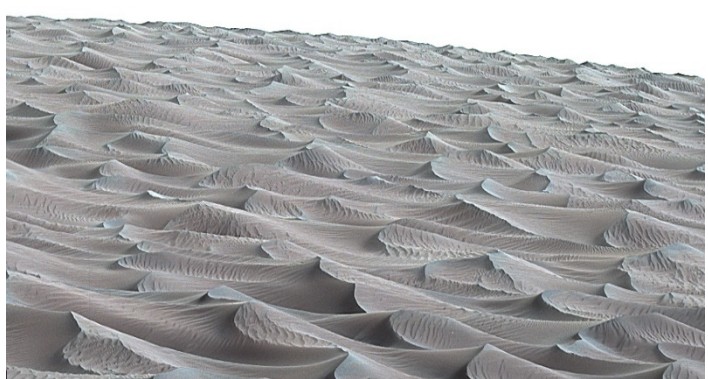
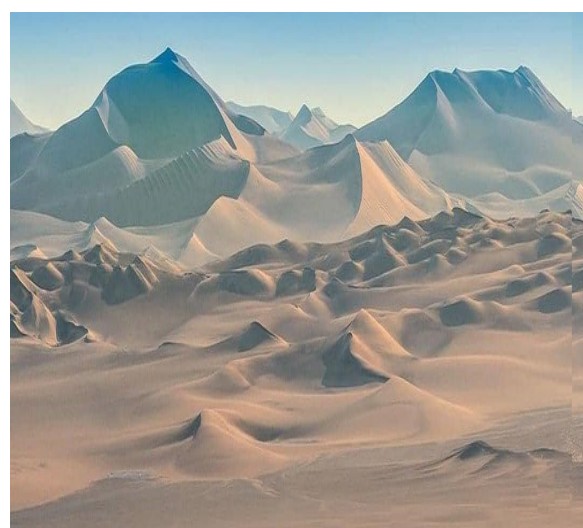

**Figure 10.** Sand dunes in LDWHS and Mars (source: research findings).

### 5.6. Accessibility

Space travel is not possible for all tourists [83]. Participants believe that travel to the LDWHS requires special equipment and special training, similar to travelling to Mars. According to the participants, both are challenging to reach (Figure 11).

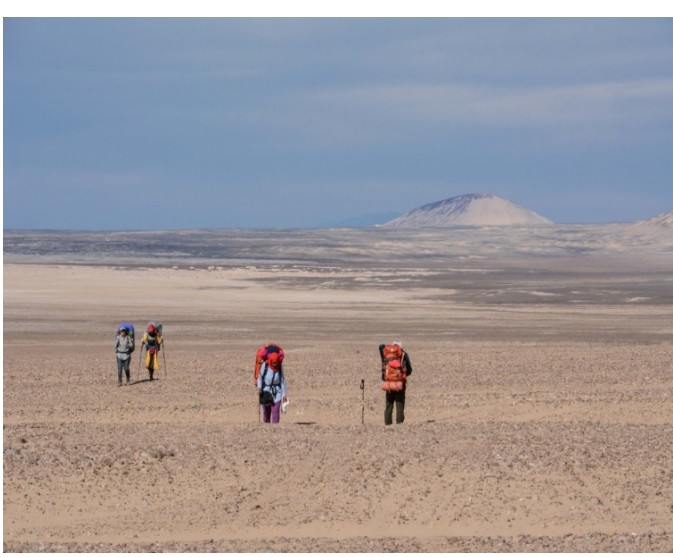
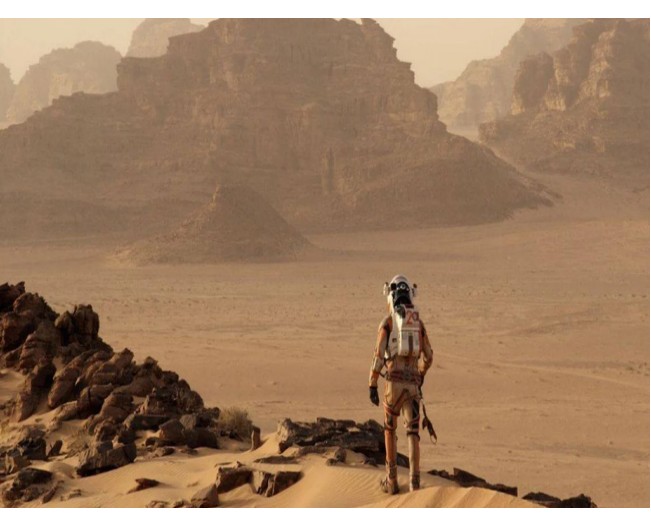

**Figure 11.** LDWHS and Mars astronauts (source: research findings).

"A trip to the Lut Desert can be a full-fledged, enjoyable, and relaxing adventure, or it can easily turn into a bitter memory. Which of the two happens depends on the preparation before you travel to the desert attractions. Tourists should be aware of the dangers of traveling to the desert in summer and winter and be

familiar with the principles and skills of rescue in the desert. Map reading skills and navigation in desert areas must also be trained," (Participant 15)

"In Iran, the LDWHS is one of the most popular and special attractions for nature and tourism, both for domestic tourists and foreign travelers. Thousands of people travel to the LDWHS areas of Iran every year, and without a doubt, without knowing how to walk in the desert, one cannot fully understand the beauties of traveling to such relaxing and stubborn areas. Lack of familiarity with the principles of desert climbing causes premature dehydration and heatstroke. Tourists are advised to study the necessary training for traveling to the desert before traveling to the LDWHS and travel to these areas with full equipment," (Participant 11)

"Although tourist tours to the LDWHS are licensed by the General Directorate of Cultural Heritage, Handicrafts, and Tourism, due to extreme heat, flooding, and the lack of traffic in parts of the LDWHS, entry to these areas is dangerous for tourists. For this purpose, tourists must have travel information and equipment for desert tourism and receive the necessary training," (Participant 1)

*5.7. Geomorphology*

However, previous studies have reported the unusual geomorphology of the LDWHS [73]. There has been little attempt to understand other surficial and shallow subsurface processes occurring in the LDWHS [67]. Moreover, the recent expeditions found a shallow subsurface water system in the LDWHS [15]. During the interview, participants exemplified the Martian mountains of Nehbandan and emphasized that the soil types in both areas have many similarities in terms of color. For example, one of the participants who traveled to the Atacama Desert believed that the Atacama Desert and the LDWHS are Mars alternatives (Figure 12).

"Strange silence and clear sky, the farther we went from the village, the more my fear increased because I knew we were going to be in the heart of the desert for hours. The whole scene was exactly what I saw in the movies. As the sun went down, the sun hid behind the dunes and gave way to the stars, and the stars slowly began to flicker in the sky. And I was always whispering to myself, am I on earth?" (Participant 12, moonlight hike tourists)

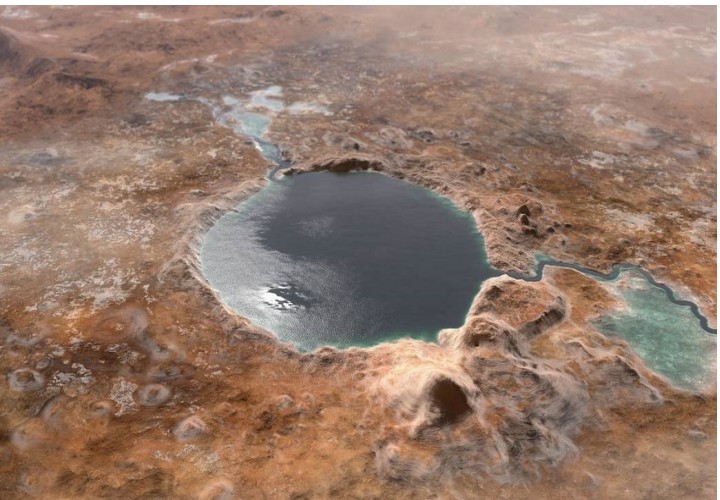 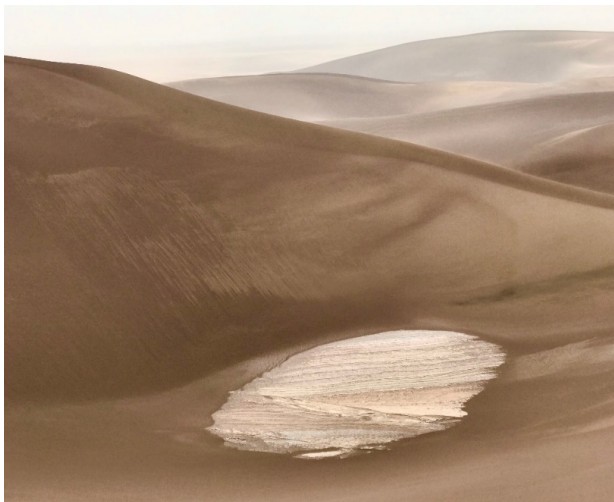

**Figure 12.** LDWHS and Mars (Jezero Crater was a lake in Mars' ancient past) [84].

"Before the trip to the LDWHS, I thought that only Atacama Desert could look like Mars in terms of climate and geomorphology. During my many trips to the LDWHS, I realized that the LDWHS in Iran is very similar to what I had found in my research on the LDWHS due to the existence of phenomena such as meteorites, soil color, and Martian mountains. I passed this information on to my colleagues through sampling," (Participant 4)

"In dry sandy deserts such as the LDWHS, which have less water erosion and more wind erosion, meteorites can be better identified and discovered. In recent years, a group of Russian geologists, in an exploratory trip to the LDWHS with Iranian researchers, have discovered 13 kg of meteorite-like matter. Based on this, it can be said that the LDWHS has a high potential for finding extraterrestrial objects," (Participant 6)

## 6. Conclusions

While a number of studies have mentioned that the Lut Desert World Heritage Site has various geological and geographical capabilities, the relevant literature throws little or no light on how this desert has positioned itself competitively in the growing global market of tourism. For this purpose, the present study was an attempt to show the comprehensive capabilities of the Lut Desert World Heritage Site with a new perspective on tourism management for its desirability to replace travel to Mars. In addition, in this study, we tried to attract the attention of researchers, especially in the field of tourism. The purpose of the present study was to introduce the subject of sustainable ecotourism and the desirability of traveling, specifically, the integration of tourism within a global system of development and impact so that this area is preserved for the sustainable development of tourism in the future of the new millennium. With the rise of environmental awareness around the world, more and more tourists are becoming conscious of the impact they cause on the LDWHS. Moreover, given the globalization of the LDWHS, tourists should be aware that the LDWHS's environment can no longer be dealt with as before and that as in the past, leisure activities are harmful to the LDWHS. From the perspective of the research method, in this research, an attempt was made to present a new model according to tourism needs in the future. The model used in this study is expected to pave the way for future research and help tourism professionals and researchers to better plan tourism management. This study concludes that the LDWHS's leading desirability as a tourism destination can be attributed to its long-term and integrated strategic planning headed by its government and policymakers getting involved in world heritage, with the impactful engagement of stakeholders across different competitiveness pillars. It is found that the prosperity of LDWHS tourism can be used in an innovative way by creating a sustainable tourism value proposition that attracts investors while also generating economic livelihood for local people. The sustainable development of LDWHS tourism in addition to its preservation can be a stimulus for the development of eastern Iran. Sustainable ecotourism also pays attention to human footprints in nature, and it can preserve the Lut Desert World Heritage Site for the future. Although the study highlighted and identified novel and key potentials of LDWHS sites for its desirability to replace travel to Mars in the form of themes and sub-themes, future studies could focus on and examine each of these novel tourism capabilities in the LDWHS. It is also recommended that the best places for such activities in the LDWHS potential areas be identified. The results show that local communities live in the form of villages and oases around most of the LDWHS potential areas.

This study concludes that the LDWHS's desirability as a tourism destination can contribute to its long-term and integrated strategic planning, which is conducted with the effective participation of stakeholders in various pillars as well as the cooperation of government and policymakers involved in world heritage. It is found that the prosperity of LDWHS tourism can be used in an innovative way as a value proposition to attract sustainable tourism investors on one hand, while encouraging the creation of economic livelihood for local people on the other. The sustainable development of LDWHS tourism

in addition to its preservation can be a stimulus for the development of eastern Iran. Sustainable ecotourism also pays attention to human footprints in nature and it can preserve the Lut Desert World Heritage Site for the future. Travel to the LDWHS is not available to everyone, especially travel to the sites discussed in this research, travel for the elderly, and travel for those who have respiratory problems and need health care. According to the slogan of the UNWTO, tourism is for all. Since this research team works in World Heritage sites in Iran, for the practical output of this research, researchers have made correspondence to show the attractions of the LDWHS and other World Heritage sites in the context of virtual and online travel for everyone. By introducing these tourist attractions to rural desert areas, this study emphasized the importance of reviving tourism and tourism economies for locals in these areas, although future studies could explore other rural areas where tourism drives economies. It is believed that the method we used in this research can be used in other crises affecting ecotourism. Efforts should be made to reinforce sustainable ecotourism in future research and to scientifically demonstrate the disadvantages of mass tourism, such as the destruction of the environment and pristine areas, with field observations and examples. Although this research was in a specific area, in future research, identifying similar phenomena in other areas can lead to their sustainability, which is in line with the goals of sustainable development and sustainable tourism. Future research can also focus on identifying and designing tourism management guidelines and policies to maintain the health of tourists in times of crisis. The complementing psychological research in tourism management in times of crisis, such as with COVID-19, is felt more strongly than ever, so future research should try to identify strategies and scenarios for improving the mental and physical health of tourists in times of crisis. Subsequently, although recent studies and researchers in this study emphasize the high value and credibility of qualitative studies in tourism research during crisis eras such as COVID-19, future research can examine the components extracted from the research based on quantitative studies. In summary, all stakeholders, including us as researchers, have a duty of big liability: to help redirect and restart tourism towards surely sustainable and resilient tourism that is fit for the COVID-19 era and a future that is steadily changing and is full of new crises.

### 6.1. Limitations

Due to the special climate conditions, any research trip in the Lut Desert is greatly affected by time constraints because it is impossible to travel to many areas of the desert during the hot months of the year. The current research is the result of a three-year time process, some stages of which were carried out at a temperature of 55 degrees. On the other hand, the difficulty of accessing some Martian sites has been one of the most important limitations of the current research.

### 6.2. Recommendations and Future Research

Moonlight hike tours in the LDWHS are considered as one of the serious outputs of this research. Most of the participants admitted that they had participated in these tours and consider the travel time in these tours to be as good as possible to see the attractions of the LDWHS and compare it with Mars. Moonlight hike tours can be the source and beginning of extensive studies on LDWHS tourism. It is suggested that in future research, sustainable tourism strategies be pursued in the form of entrepreneurship to strengthen the economy of the local community in the LDWHS. Efforts should be made to scientifically demonstrate the disadvantages of mass tourism for the destruction of the environment and pristine areas with field observations and examples from the LDWHS. Moreover, the study of sustainable ecotourism in future research should be investigated in more depth. In order to restart tourism in the LDWHS during and after the coronavirus epidemic, it is suggested that further studies focus on the future state of tourism in the LDWHS and show a platform for overcoming the pandemic and restarting tourism in future research. The desirability of travel in this study can be introduced as an alternative destination. Alternative tourism includes non-gathering places and is determined by dedicated tours

(such as the moonlight hike tours in this study) and tourists' desire to experience specific environments. It is suggested that in future research, alternative destinations be discussed based on the desirability of travel in tourism studies. It seems that because of COVID-19, local people, businesses, and societies try to find new economic methods to strengthen the local economy and will seek to change the way tourism and hospitality are managed. According to the findings of this section, the presence of the interesting phenomena in the Lut Desert World Heritage Site can strengthen the local economy of the host communities. In line with these results, we also believe that lessons from COVID-19 can prepare global tourism for the economic transformation needed. Moreover, recent research on COVID-19's effects on the tourism economy emphasizes that the ways of doing tourism business need to change and recommends that "tourism destinations will have to rethink their business as usual approaches going into the future".

**Author Contributions:** Conceptualization, A.G. and H.M.; Methodology, A.G.; Software, A.G., A.Z., H.M. and F.A.A.; Validation, A.Z. and K.Z.; Formal analysis, H.M., F.A.A. and K.Z.; Investigation, F.A.A.; Resources, K.Z.; Writing—original draft, A.G. and H.M.; Writing—review & editing, A.G., A.Z., H.M., K.Z. and L.D.D.; Visualization, A.Z., F.A.A. and L.D.D.; Supervision, K.Z. and L.D.D.; Project administration, H.M.; Funding acquisition, K.Z. and L.D.D. All authors have read and agreed to the published version of the manuscript.

**Funding:** This research was supported by the Hungarian University of Agriculture and Life Sciences and the Doctoral School of Economic and Regional Sciences (MATE), Hungary.

**Institutional Review Board Statement:** Not applicable.

**Informed Consent Statement:** Not applicable.

**Data Availability Statement:** Data developed in this study will be made available upon request to the corresponding authors.

**Acknowledgments:** This research was carried out with the official permission of the director of the Lut Desert World Heritage Site, Mehran Maghsoudi with the number (BL99006). Hereby, the research team expresses its gratitude to him and the following people who participated in the data collection and identification of sites process: 1. Shahdad Kalantari (Lut Desert special tour guide and founder of the moonlight hike tours programs). 2. Ehsan Etemadinia (photographer of the Lut Desert World Heritage site). 3. Zahra Rezaei Malakooti (responsible for the Lut base in South Khorasan Province).

**Conflicts of Interest:** The authors declare no conflict of interest.

## Appendix A. Research Questions

1. What is the difference between moonlight hike tours and other travel packages?
2. What is the main reason for choosing a moonlight hike tour by you?
3. How have you felt about participating in the tour so far?
4. Tour officials believe the package will help preserve the Lut Desert and promote sustainable tourism in the region. What do you think?
5. Have you ever felt that you are in a place other than Earth by being in the desert of Lut? For example, another planet (Moon or Mars)?
6. If so, in which part of the Lut Desert have you ever experienced this feeling?
7. What do you think of the Lut Desert Martian sites?
8. Which feature of the introduced sites evokes the feeling of being on Mars the most?
9. Do Martian sites have the potential to be introduced as a new branch of tourism for the Lut Desert?
10. What are the main similarities between the Lut Desert and Mars, given the images, videos, and descriptions provided at the beginning of the tour? (Before the tour, tour operators will hold an orientation class for tourists and explain the similarities between the Lut Desert and Mars, and videos and images from the surface of Mars will be shown to tourists.

## Appendix B. LDWHS Landforms

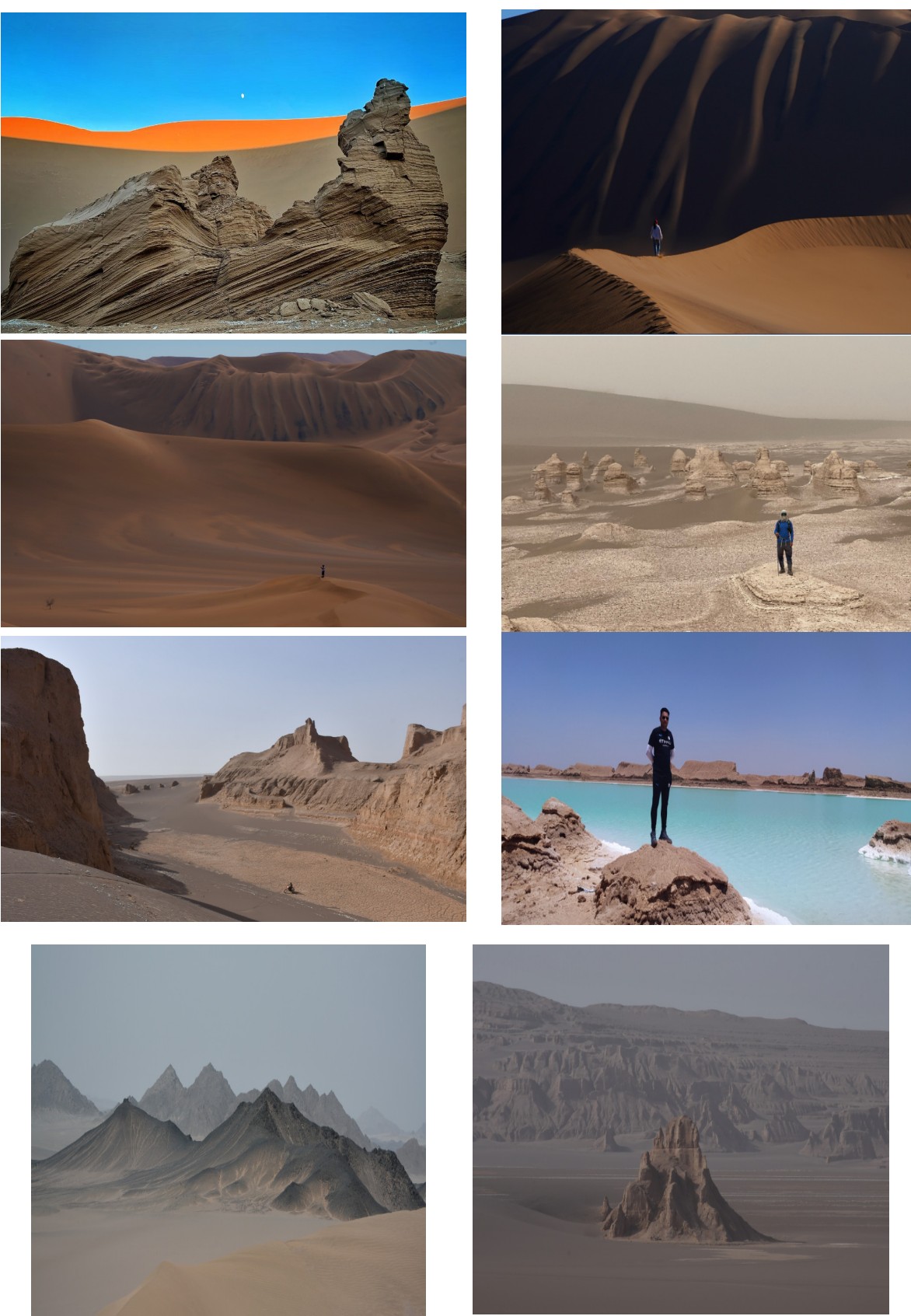

**Figure A1.** LDWHS landforms (source: research findings).

## Appendix C. LDWHS Moonlight Hike Tour Class

Moonlight hike tour preparation class. Familiarity with the LDWHS and its similarities with Mars. A map of the Lut Desert is printed on the tour leader's T-shirt. The leader's most important emphasis is on the sustainability principles in Lut tourism.

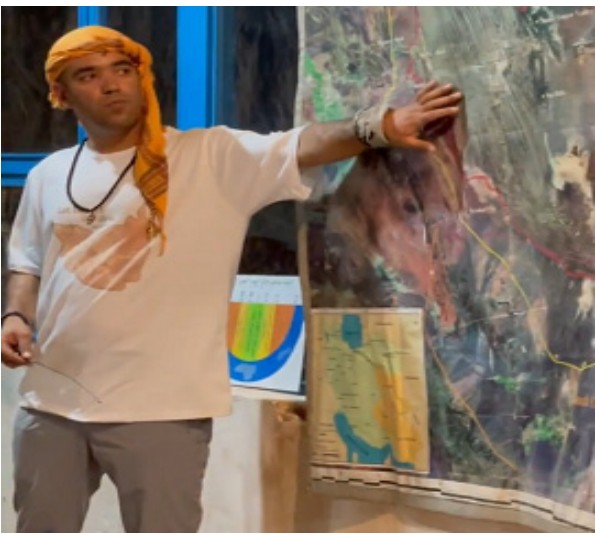

**Figure A2.** Moonlight hike tour preparation class (researchers, 2023).

Figure A3 shows the path from the Shahdad Kaluts to the crater of Gandum Baryan volcano:

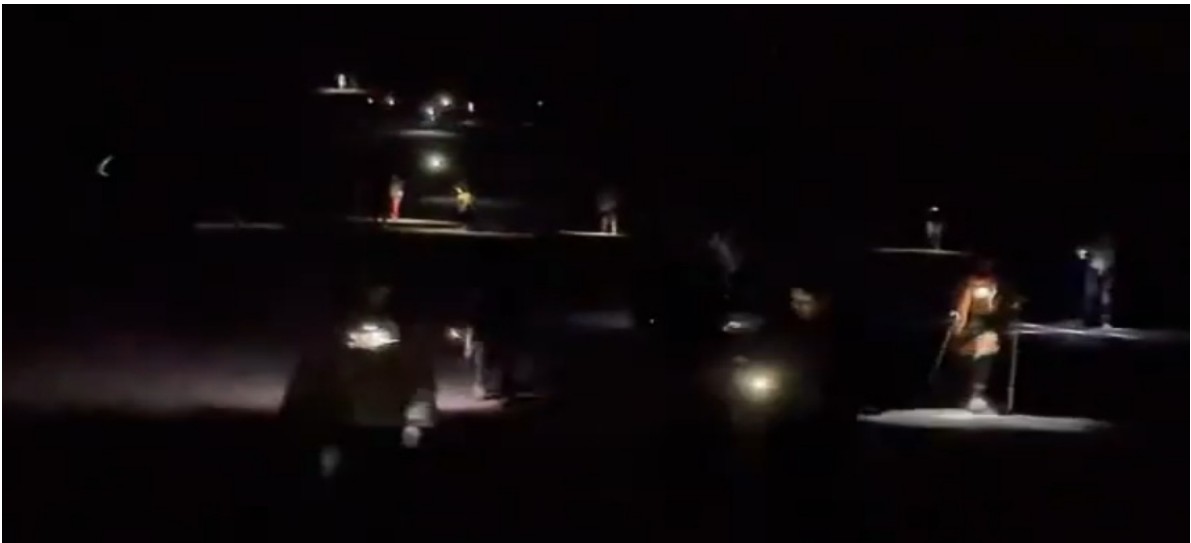

**Figure A3.** Moonlight hike tourists on the way to Martian sites; they should be at the Martian site at sunrise (researchers, 2023).

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
