# Peer review of "Travel to Mars-like Places on Earth: A New Branch of Sustainable Ecotourism in Lut Desert World Heritage Site, Iran"

_sustainability, doi:10.3390/su15129677_

Round 1

Reviewer 1 Report

This is a very useful article. Thank you for this. It is well designed. The paper contains new and significant information adequate to justify publication. I however made a few comments mainly on the methodology section and the discussion section. Hope the authors will find them useful.

1. how do you calculate the number of research samples in Table 1?

2. Is there any ethic approval for some pictures and questionnaires in the study?

3. Discussion part should be combined with literature review.  In discussion section, the literature and the findings are engaged with simultaneously, the author needs to succinctly demonstrate how the literature fits into the findings, and mention emphatically how these are juxtaposed with each other.

4. Research limitations should be a part of Conclusion section, please remove it from Discussion section.

Author Response

Hello dear reviewers:

Many thanks for your time and effort toward the promotion of our manuscript. We have eagerly read the comments and believe they improved the manuscript. In the form below, your valuable comments and our edits to the sections in question are highlighted. It is hoped that the manuscript will now be sufficient for publication.

Authors

Sincerely yours

Dear reviewer many thanks for your kind words. We will try to improve the manuscript according to your opinion

Dear reviewer, According to the text, the statistical sample consists of experts and official guides of Lut Desert, as well as tourists who participated in Lut Desert moonlight tours. Due to the limited size of the statistical sample, all people participated in the interview process.

Dear reviewer: Many thanks for your inside. All images belong to the researchers and have not been copied from anywhere. Images are provided for the current manuscript.

Many thanks for your valuable comment. Edited.

Many thanks for your valuable comment. Edited.

Reviewer 2 Report

The title of the article is in line with its content. However, the summary part of the article could better reflect the findings obtained from the study. In the abstract, the sample of the study should be given more clearly. No information was given about ecotourism and sustainable ecotourism in the introduction of the research, but sustainable ecotourism seems to be the subject of the article. This deficiency must be corrected. In addition, the summary of the literature includes the title postmodern ecotourism. There is a need for literature knowledge on postmodernism and postmodern ecotourism under this heading. When the article is read in general, it is understood that the study was conducted during the Covid-19 pandemic process. Therefore, there are findings and suggestions related to the pandemic process. However, no information was given about the schedule of the study in the introduction or method section. In addition, no emphasis was given to the Covid-19 pandemic in the introduction section.

The method section of the study is well written. There is no information about the research population and how the sample was determined and which method was used. There is no information about how the 13 people were selected. In addition, no information was given about whether the sample size was statistically sufficient to reflect the population. There is no information on how the research questions in the appendix were written and tested and found appropriate. If an expert opinion has been obtained on the subject, it must be included in the method section.

Author Response

Dear reviewer: Many thanks for your inside. Edited according to your comments.

Dear reviewer: Many thanks for your insight. The statistical sample of the research consists of 18 people, including experts and also members of a tour who have visited Martian sites. Due to the difficulty of accessing the studied sites and the small number of people who managed to visit these sites, it is not possible to increase the sample.

Reviewer 3 Report

The methodology is well presented as well as the results and discussion.

The paper reflects widely on the role of ecotourism in postmodernism and it is therefore recommended to include a reflection on what is meant by postmodernism.

It is recommended to check misplaced punctuation marks.

Author Response

Many thanks for your kind words.

It means sustainable ecotourism, which corresponds to the approaches of postmodern theorists. Therefore, postmodern was replaced by the term sustainability.

Dear reviewer checked according to your comment.

Reviewer 4 Report

In point 3.1 Study area, it is important to present brief information about the tourist demand for the destination - number of visitors for the last years

page 7, line 241 - repetition of "was used"

page 9, line 262 - unclear sentence

Author Response

Many thanks for your insight. Added according to your valuable comment.

Many thanks for your valuable comment.

Many thanks for your valuable comment. Edited.

Reviewer 5 Report

The article is very well presented. The research was well conducted. It is only recommended that the authors review the references used. For example, the study by (Mac Donald et al., 2020), does not appear in the list of references. 

Good work.

The quality of English language is acceptable for a scientific paper. 

Author Response

Hello dear reviewers:

Many thanks for your time and effort toward the promotion of our manuscript. We have eagerly read the comments and believe they improved the manuscript. In the form below, your valuable comments and our edits to the sections in question are highlighted. It is hoped that the manuscript will now be sufficient for publication.

Authors

Sincerely yours

Dear reviewer checked according to your comment.
